# High Dosage of Botulinum Toxin Type A in Adult Subjects with Spasticity Following Acquired Central Nervous System Damage: Where Are We at?

**DOI:** 10.3390/toxins12050315

**Published:** 2020-05-10

**Authors:** Domenico Intiso, Valentina Simone, Michelangelo Bartolo, Andrea Santamato, Maurizio Ranieri, Maria Teresa Gatta, Filomena Di Rienzo

**Affiliations:** 1Unit of Neuro-Rehabilitation Unit and Rehabilitation Medicine, IRCCS ‘Casa Sollievo della Sofferenza’, Viale dei Cappuccini 1, San Giovanni Rotondo, 71013 Foggia, Italy; v.simone@operapadrepio.it (V.S.); mt.gatta@operapadrepio.it (M.T.G.); f.dirienzo@operapadrepio.it (F.D.R); 2Department of Rehabilitation, Neurorehabilitation Unit, HABILITA Zingonia, Ciserano, 24040 Bergamo, Italy; bartolomichelangelo@gmail.com; 3Department of Physical Medicine and Rehabilitation, OO.RR Hospital, University of Foggia, 71122 Foggia, Italy; andrea.santamato@unifg.it (A.S.); maurizio.ranieri@uniba.it (M.R)

**Keywords:** botulinum toxin, spasticity, high doses, rehabilitation

## Abstract

Spasticity is a common disabling disorder in adult subjects suffering from stroke, brain injury, multiple sclerosis (MS) and spinal cord injury (SCI). Spasticity may be a disabling symptom in people during rehabilitation and botulinum toxin type A (BTX-A) has become the first-line therapy for the local form. High BTX-A doses are often used in clinical practice. Advantages and limitations are debated and the evidence is unclear. Therefore, we analysed the efficacy, safety and evidence for BTX-A high doses. Studies published from January 1989 to February 2020 were retrieved from MEDLINE/PubMed, Embase, Cochrane Central Register. Only obabotulinumtoxinA (obaBTX-A), onabotulinumtoxinA (onaBTX-A), and incobotulinumtoxinA (incoBTX-A) were considered. The term “high dosage” indicated ≥ 600 U. Thirteen studies met the inclusion criteria. Studies had variable method designs, sample sizes and aims, with only two randomised controlled trials. IncoBTX-A and onaBTX-A were injected in three and eight studies, respectively. BTX-A high doses were used predominantly in treating post-stroke spasticity. No studies were retrieved regarding treating spasticity in MS and SCI. Dosage of BTX-A up to 840 U resulted efficacious and safety without no serious adverse events (AEs). Evidence is insufficient to recommend high BTX-A use in clinical practice, but in selected patients, the benefits of high dose BTX-A may be clinically acceptable.

## 1. Introduction

Spasticity is a common disabling disorder in adult subjects suffering from upper motor neuron syndrome (UMNS) and generally develops after lesions of the central nervous system (CNS) such as those caused by stroke, brain injury, spinal cord injury (SCI), multiple sclerosis (MS) and cerebral palsy (CP). The physiopathology and muscular changes are not fully clear, though many definitions have been proposed; Lance’s definition remains the most diffuse and widely accepted. Accordingly, spasticity is a motor disorder characterised by a velocity-dependent increased resistance to passive limb movement with increased muscle tone and hyperactive reflexes [1]. On the other hand, a key role has been attributed to muscle tissue change, consequently, it is defined as pathological neuromuscular activation and hyper-resistance around joints due to modification of visco-elastic muscle components [2]. Regardless of the pathophysiological mechanisms underlying this motor and muscular tone condition, spasticity may be a disabling and challenging symptom in people during the rehabilitation process and recovery. The disorder can hamper the functional outcome and contributes to the loss of motor dexterity and ability by promoting a persistent abnormal posture, muscular-tendon contractures, bone deformity and a poor quality of life. Therefore, to reduce its negative impact and to improve functional limitations, several therapeutic strategies, including non-pharmacological and pharmacological strategies, have been proposed. Spasticity can occur among the spectrum of UMNS symptoms that include weakness, fatigue, spasms and clonus and, thus, the clinical picture can be variable depending on the sites involved and extension or neuronal tissue damage. Moreover, spasticity can be segmental, involving limited sites of the body, or generalised, involving both the upper and lower limbs. Several medical and surgical therapeutic strategies have been proposed for generalised spasticity. Conversely, botulinum toxin type A (BTX-A), by inhibiting the release of pre-synaptic acetylcholine at neuromuscular junctions, has become the first-line therapy for treating focal or segmental spasticity. Current clinical recommendations for muscle-specific dosing, sites of injection, dosage and dilution as well as the formulation of BTX and adjunct treatments to boost effects are based on the injector’s decision [3]. Although widely debated, the BTX-A doses for muscle and the total injection dosage for sessions remain unclear [4]. In this regard, the European Consensus conference recommended that the BTX dosage should not exceed 600 U and 1500 U for injection sessions when using onaBTX-A and aboBTX-A, respectively [5]. Likewise, consensuses have been agreed about the muscles and sites of injection [6]. However, the BTX dosage is largely titrated by the practitioner based on the individual patient response and spastic pictures of patients in order to obtain specific functional goals [7]. The global dosage injected per muscle or multiple muscle groups has progressively increased over time and higher BTX dosages than that recommended are often used in clinical practice and real-world use [7,8], particularly in post-stroke spasticity. Furthermore, experts and patients have highlighted a desire for greater flexibility in treatment options, with more dosages available other than those currently approved [8,9,10]. The reasons this therapeutic approach is generally justified are its ability to reduce severe spasticity and to give a more tailored treatment in patients with multi-level spastic muscles [11,12]. The risk of diffusion and occurrence of severe side effects represent important concerns regarding the injection of high doses. Several studies have studies the injection of high-dose BTX in treating spasticity, but the indications, advantages and limitations are the object of debate, and the evidence from published studies is unclear. Herein, we will discuss the efficacy, safety and evidence for the use of high dose BTX-A in treating spasticity following common neurological disorders due to CNS damage in adult patients requiring rehabilitation.

### BTX-A

Botulinum neurotoxin (BTX), so called botulinum toxin, is a protein synthesized by the Gram-negative anaerobic bacteria Clostridium botulinum. It occurs in the form of seven serologically distinct types, each of which comprises numerous isoforms. Most of the research available in the medical literature regards the neurotoxin type A (BTX-A). The classic site of the impact of the toxin is the protein complexes of the presynaptic membrane and synaptic vesicles, implementing the acetylcholine exocytosis. Type A botulinum toxin can be effective in the treatment of drug-resistant migraine. It is also widely used in aesthetic medicine for the correction of age-related changes in muscle tension, but also otherwise for the treatment of painful bladder, chronic myalgia, blepharospasm and some kinds of neuralgia. In recent decades, several BTX-A formulations have been manufactured by Asian and Brazilian pharmaceutical industries and licensed by the Health Agency. Hereby, the following formulations mainly used in Europe and USA will be considered: abobotulinumtoxinA, Ipsen Biopharm Ltd (ObaBTX-A); onabotulinumtoxinA, Allergan Inc (OnaBTX-A) and incobotulinumtoxinA, Merz Pharmaceuticals GmbH (IncoBTX-A), commercially named Dysport®, Botox® and Xeomin®, respectively. OnaBTX-A and obaBTX-A are composed of 150 kD active neurotoxin and non-toxic accessory proteins, whereas incoBTX-A contains only the 150 kD neurotoxin [13]. The agents are not interchangeable and there are no validated conversion ratios. Since incoBTX-A has a similar clinical efficacy and adverse event profile as onaBTX-A, it is suggested that the dose ratio should be 1:1 or 1:2; whereas a variable dose ratio from 1:3 to 1:5 has been proposed for the conversion between ona- and aboBTX-A [14,15,16].

In the present paper, the term “high dosage” is considered as neurotoxin doses injected in a single session higher than 600 U for onaBTX-A and incoBTX-A and higher than 1500 U for aboBTX-A toxins.

## 2. Results

### 2.1. Stroke

Stroke is a leading cause of disability worldwide and spasticity occurs with a range from 18% to 43% [17,18,19]. Thus far, post-stroke spasticity (PSS) is widely treated by BTX-A and several health drug agencies, including the U.S. Food and Drug Administration (FDA) and European Regulatory Agencies, have licensed BTX types and formulations as well as doses to inject for upper (ULS) and lower limb spasticity (LLS). In this regard, in USA and Europe, onaBTX-A (Botox), obaBTX-A (Dysport) and incoBTX-A (Xeomin) have been approved to treat ULS, while only onaBTX-A (Botox) and obaBTX-A (Dysport) have been approved to treat LLS. Accordingly, doses up to 400 U for onaBTX-A and incoBTX-A, and 500–1000 U for obaBTX-A, have been approved for treating ULS. Likewise, doses of 300–400 U for Botox, and up 1500 U for Dysport, have been approved for LLS. There is now a well-established body of evidence demonstrating the safety and efficacy of BTX-A in reducing spasticity both in the upper and the lower limb [20,21,22,23]. A multitude of studies and several guidelines have been published reporting recommendations and evidence classes for muscles, dosage and dose intervals for injection when using BTX-A to treat PSS [6]. Although guidelines in Europe allow for doses up to 600 U of onaBTX-A in adults with spasticity [5], many studies have been published about the injection of higher doses of BTX-A in subjects with PSS, and three reviews have focused this topic [24,25,26]. The literature search identified 1730 citations, but only 13 studies met the inclusion criteria The studies varied in method design, had small or mixed samples and only two studies included a randomised controlled design [27,28]. Of these, the effect of high doses of BTX-A in reducing spasticity were investigated only in one trial [27]. Similarly, the aim of investigations was variable and only a few studies focused on the effects and safety of higher doses of BTX-A than recommended. IncoBTX-A and onaBTX-A were predominantly used. BTX-A formulation alone or associated with other treatments were injected in three and eight studies for onaBTX-A and incoBTX-A, respectively. Two studies used both onaBTX-A and incoBTX-A formulations [28,29]. High-dosage aboBTX-A was described in only one study. Below, we describe all studies regarding the injection of BTX-A high dosage that enrolled homogeneous samples of patients with stroke or mixed samples that included subjects suffering from this disease. Studies with mixed samples not including stroke or not reporting causes of spasticity are discussed in different paragraphs of the present paper. The studies have been shared for the authorised types of BTX-A.

Table 1 shows all reported studies, apart from case reports or those describing single subjects.

#### 2.1.1. AbobotulinumtoxinA (Dysport)

Only one study has been published reporting high dosage aboBTX-A in subjects with PSS. Six patients were efficaciously treated by 2000 U for LLS [30] and only one subject demonstrated side effects, consisting of bladder paresis.

#### 2.1.2. OnabotulinumtoxinA (Botox)

High dosage of onaBTX-A injections has been reported in six studies, but only three were included in the present review since three of these studies were case reports describing adverse events [31,32,33]. All trials investigated the effectiveness and safety of high dose onaBTX-A in reducing spasticity of post-stroke subjects, even if different method designs were used [17,27,34]. Of these, two studies used a retrospective design. The former reported a cohort of 26 post-stroke subjects who were injected with a mean dose of 676.9 ± 86.3 U of onaBTX-A. Twenty-three patients were treated at both upper and lower limbs and 13 patients received 700 U. A significant reduction in spasticity was observed (*p* < 0.0001), and no adverse events occurred [34]. The latter study included a mixed sample of patients [12] with dystonia and spasticity to evaluate the safety of onaBTX-A injections at dosages higher than 400 U. The study enrolled 68 patients and, of these, 24 had spasticity following stroke, brain injury and cerebral palsy. A mean total dose of 501 ± 46 U (range 425–800 U) was injected. The mean follow-up period was 23 months (range 3–86). All patients reported a benefit after the first treatment (8.8 ± 3.1 weeks). However, 13 patients (19%) reported AEs at one year, and 7 (10%) at the last follow-up. Severe dysphagia or major AEs requiring hospitalisation or additional interventions were not observed. The authors suggested that onaBTX-A dosages >400 U in a single session could be safely injected, were efficacious and had lasting benefit. The other study was a randomised, double-blind study and evaluated three doses of onaBTX-A in 45 subjects randomised to 3 groups of 15 patients to treat spasticity in the foot. Each group received 166.7 ± 30.9 U, 321.7 ± 92 U and 540 ± 124.2 U (high dose) mean doses of onaBTX-A, respectively. All groups showed a significant improvement in spasticity, but patients in the high dose group showed a greater and longer-lasting decrease in strength in both the injected and the non-injected muscle, that in some patients endured for more than 4 weeks [27]. In total, 46 subjects were injected by high onaBTX-A doses excluding those (24 patients) treated by Chiu et al., since in these patients spasticity aetiology was not reported.

#### 2.1.3. IncobotulinumtoxinA (Xeomin)

Five studies concerned the use of high doses of incoBTX-A. Of these, three studies enrolled only post-stroke patients and two included mixed samples. Of the studies that included only post-stroke subjects, two studies had an open-label, prospective method design and were performed by Santamato et al. [35,36]. Both concerned the same sample of patients. The former investigated the effect of incoBTX-A doses up 840 U (range from 750 to 840 U) injected in muscles of both UL and LL in the same session to reduce multi-level spasticity. The upper limb was injected by a maximum dosage of 540 U, whereas a dosage of 340 U was administered into the lower limbs (range 250–340 U). Over 10 days, subjects underwent stretching exercises of the muscles after receiving injections. A significant decrease in spasticity without AEs was observed after 30 and 90 days from the treatment (*p* < 0.05). Furthermore, improvements in functional disability, spasticity-related pain and muscle tone were detected [35]. The latter study reported the effect in this population treated for 2 years by high doses. In a two-year follow-up, repeated high doses of incoBTX-A, administered for eight sets of injections were safe and efficacious in reducing UL and LL spasticity without generalised AEs [36]. The third study, with a homogenous sample of stroke subjects, investigated the effect of incoBTX-A > 600 U on the autonomic nervous system (ANS) by measuring changes in heart rhythm and enrolled 11 (5 M, 6 F, mean age 59.55 ± 12.8 years) patients. Subjects were injected with 12U/kg of incoBTX-A (range 600–800; mean dosage 677 ± 69.3 U). Two recording ECGs were performed, one before neurotoxin injection and the second one 10 days after the treatment. No differences were observed in parameters of heart rate variability [37]. The remaining two studies included subjects with spasticity following several diseases affecting the central nervous system (CNS). Of these, one was a large prospective, multicentre, single arm, open label, dose-titration study (TOWER study). The trial enrolled 155 patients and of these, 132 had PSS. Although a mixed sample, given that most of the subjects had stroke we preferred to describe it in this section. The primary objective was to investigate the safety of high doses of incoBTX-A and the investigators’ global assessment of tolerability. One hundred and fifty-five post-stroke subjects received escalating doses of 400, 600, and 800 U of incoA in the same body site. One-hundred thirty and seven patients (88.4%) completed the study and 82.9% (116/140) received 800 U. With escalating total doses, a higher number of spasticity patterns was successfully treated. IncoBTX-A dose escalation from 400 U up to 800 U increased improvements of muscle tone, goal attainment and global efficacy. Transient adverse events occurred for each dose group and there was no increased incidence of adverse effects with increasing doses. The most frequent side effects overall were falls (7.7%), nasopharyngitis, arthralgia and diarrhoea (6.5% each) [38]. Globally, five subjects discontinued treatment due to AEs, but none of these belonged to subjects receiving 800 U. No patients developed secondary nonresponses due to neutralising antibodies. The last study by Ianieri et al. reported the effect of escalating doses up 1000 U of incoBTX-A to reduce spasticity according to the individual patient’s features and needs. The doses were chosen depending on the severity of spasticity. One hundred and twenty subjects suffering from spasticity due to several pathologies were retrospectively analysed and 58 patients received high incoBTX-A doses from 400 U to 1000 U. The authors individualised treatment for three groups of patients who were injected by ≤400 U; 400–700 U and 700–1000 U, respectively. The group treated by incoBTX-A > 700 U was injected by progressively increasing dosages, with a mean dose between 775.65 ± 30.45 and 986.65 ± 13.67 U. Transient general weakness was observed in 4% of subjects [39].

It was not possible to calculate the global number of treated patients, since one study with mixed samples did not report aetiologies of spasticity [39]; however, it was estimated that about 200 patients with PSS were injected with high dose incoBTX-A.

#### 2.1.4. OnaA or IncoA

Two studies used onaBTX-A or incoBTX-A. The former by Baricich et al. compared the effect of high doses of onaBTX-A and incoBTX-A on the cardiovascular activity of the ANS in chronic hemiplegic spastic stroke survivors [28]. The study used a single randomised controlled crossover method design and enrolled a very small sample. Ten patients (mean age 69 ± 10.5) were randomised into 2 groups of 5 subjects and each group received onaBTX-A or incoBTX-A doses higher 600 U. The onaBTX-A group was injected by a mean dosage of 670 ± 83.67 U and the incoBTX-A group by a mean dosage of 660 ± 89.44 U (doses below 12 units/Kg). No effect was observed [28]. The latter by Kirshblum S et al. [29] was a retrospective study to determine differences in risk of AEs when using doses higher that 600 U of onaBTX-A or incoBTX-A as compared to lower doses. The study investigated a large sample of 342 subjects suffering from dystonia and spasticity, but only 42 patients received doses higher than 600 U. The aetiology of spasticity was not reported. In this group, doses higher 600 U were found to increase the rate of adverse effects.

### 2.2. Brain Injury

Spasticity frequently occurs after brain injury and many studies and meta-analysis have been reported about the use of BTX-A in treating UL and LL in this disorder [40,41,42]. Although subjects suffering from spasticity following brain injury may show complex features and multi-level muscle involvement, few studies have investigated the efficacy and safety of high dose BTX-A. The literature research produced 246 citations, but only 4 studies were included [12,38,39,43]. The studies had variable method designs and enrolled mixed samples including patients with spasticity following several disorders affecting the CNS, including brain injury, stroke and cerebral palsy (CP). Globally, 27 subjects were injected by high dose BTX-A, including onaBTX-A or incoBTX-A formulations. Of these, 16 (72.3%) subjects were enrolled in the study by Intiso et al. and 11 (7.1%) were enrolled in the study by Wissel et al. (which has been described in this manuscript under the stroke subheading). Intiso et al. evaluated the effect of incoBTX-A doses higher 700 U in 22 patients (12 M, 10 F; mean age 38.1±13.7). The sample included 16 subjects with brain injury and 6 with CP. All subjects with hemiparesis received neurotoxin injections in both the upper and lower limbs in the same session, with doses ranging from 770 to 840 U. Significant spasticity and pain reduction was detected after BTX-A injections. Side effects were transient and consisted of hematoma in the site of injection (2 subjects) and weakness of the injected arm lasting 2 weeks (one subject) [43]. The study by Wissel et al. at investigated escalating doses of incoBTX-A. In this study, 155 subjects were enrolled, but only 11 (7.1%) suffered from traumatic brain injury [38]. The remaining two investigations did not report the number of spastic subjects according to aetiology [39] or spasticity aetiology [29]. Since these studies had mixed samples, the investigation by Ianieri et al. was described in the stroke paragraph and that by Kirshblum et al. was reported in the paragraph concerning studies with mixed samples.

### 2.3. Multiple Sclerosis

The literature search produced 246 citations, but no study investigated high doses of BTX-A in treating spasticity following MS. Patients with MS can show complex neurological features due to lesions in sensory-motor and other neuronal pathways. Spasticity is a common and troublesome symptom with a variable prevalence from 30% and 60%, and up to 80%, in some series [44,45]. It can involve local or multi-level body sites. BTX-A has long been used in the symptomatic treatment of MS patients, such as for a hyperactive bladder [46]. Likewise, studies and reviews [47,48,49] have reported BTX-A use for the treatment of spasticity, but few randomised trials have been performed. In this regard, the review by Dressler et al. 2017 [49] found only three RCTs [50,51,52], and a recent Italian consensus on spasticity treatment considered only 4 trials concerning BTX [53]. Of these, two trials were randomised double-blind, placebo-controlled studies [50,52]; one [50] with a crossover design and two single-blind randomised trials [54,55]. In these studies, doses from 500 to 1500 U and from 100 to 400 UI for aboBTX-A and onaBTX-A were injected, respectively. Interestingly, one trial investigated the effect of aboBTX-A in 74 patients with MS and spasticity affecting the hip adductor muscles of both legs [52]. Subjects were randomised to 500, 1000 and 1500 U or placebo. The optimal dose for treating hip adductor spasticity was considered 500–1000 U of aboBTX-A, but the higher dose of 1500 U was suggested to provide greater benefit and was safe without major side effects [52]. Among BTX-A formulations, aboBTX-A and onaBTX-A were predominantly injected to treat spasticity, apart from in one study with mixed samples that used incoBTX-A [39] neurotoxin. All studies showed a significant efficacy of BTX in reducing spasticity due to MS [47,48,49]. BTX-A dosages up to 2000 UI for Dysport [56,57] and up to 400 UI for Botox were injected without significant side effects. European neurological Societies have published consensuses and guidelines for treating spasticity in patients suffering from MS [53,58] and have drawn up therapeutic algorithms. Accordingly, BTX-A has limited use and has been suggested only for focal spasticity. Despite this, BTX-A was easily administered and has not been associated with adverse effects to date, and no studies have been performed enrolling only subjects with MS in order to analyse the effect of high dose BTX-A. Subjects with MS were injected with high doses of incoBTX-A or onaBTX-A in one study that enrolled mixed samples [39], but limited information can be obtained (Table 2).

### 2.4. Spinal Cord Injury

The literature search produced 350 citations, but as for MS, no study investigated high doses of BTX-A in treating spasticity due to SCI. Spasticity is a frequent disorder following SCI and it is often associated with troublesome symptoms such as pain and spams that may be resistant to common therapeutic agents and worsen daily life [59]. Complicated spasticity is difficult to manage, requiring multiform interventions and multidisciplinary teams. In this regard, combined rehabilitative and pharmacological strategies have been proposed to reduce spasticity, relieve pain and improve quality of life. BTX-A is commonly used in treating coexistent symptoms such as overactive bladder [60,61] and neuropathic pain that is refractory to common analgesic agents [62,63,64]. In post-traumatic SCI, a prevalence between 60% and 80% has been reported and 35% had severe spasticity [65,66]. Although spasticity is a common disorder following SCI, few investigations have been performed to analyse the effect of BTX-A [67,68] and no studies have investigated high dose BTX-A in reducing spasticity in subjects with SCI.

### 2.5. Studies with Mixed Samples: Dystonia and Spasticity

Four studies investigating high dose BTX-A injections enrolled subjects with dystonia and spasticity due to disorders affecting the CNS (Table 2). IncoBTX-A was predominantly used, and incoBTX-A and onaBTX-A or IncoBTX-A neurotoxins were injected in three and one studies, respectively. The primary aim of these studies was to ascertain the safety of high doses of BTX-A. Of these, 2 studies were performed by Dessler et al., 2009 [11,69]. The former evaluated the efficacy and safety of incoBTX-A in patients previously treated by onaBTX-A [69]. Two-hundred and sixty-three patients were injected and, of these, 84 had spasticity, even if the aetiology was not reported. Subjects with spasticity were injected with 450.5 ± 177.1 U and those with generalised spasticity received the highest incoBTX-A doses, with an average dose of 552.2 ± 217.1 U. The maximal incoBTX-A dose applied to a single patient was 840 U, and none of the patients experienced any motor or non-motor systemic AEs. The latter was a prospective non-interventional study that evaluated high incoBTX-A doses compared to regular doses in 100 patients with dystonia or spasticity [11]. Fifty-four subjects suffered from spasticity and received a mean dose of incoBTX-A of 612.6 ± 176.5 U (min 400, max 1200 U). The author affirmed that incoBTX-A could be safely used in doses ≥ 400 U and up to 1200 U without detectable systemic toxicity and concluded that this allows injection of more target muscles and also of higher incoBTX-A doses per target muscle, where necessary. Given that it predominantly enrolled subjects with brain injury, the study by Intiso et al. has been described in the related section above [43]. Likewise, the study by Kirshblum S et al. that injected subjects with dystonia and spasticity has been reported under the stroke subheading [29].

### 2.6. Safety and Adverse Events

Local side effects are directly associated with the injections and consist predominantly of transient pain, hematoma and swelling at the injection site, and this lasts for a few days. On the other hand, systemic AEs are related to spread of the toxin to locations distant from the site of injection; these can produce serious conditions such as a botulism-like syndrome which is characterised by dysphagia, general weakness and symptoms that can resemble botulism [70]. Local and generalised AEs have been reported by repetitive BTX A injections at recommended dosage [20,71], but the use of doses higher than recommended may increase the risk of AEs. In this regard, three studies that were not included in the present paper, described AEs induced by injections of high BTX doses. AEs included fatigue and contralateral weakness after 700 U (one subject) [31] and 800 U (one subject) of onaBTX-A injections (three subjects) [32]. The remaining paper described one subject who showed upper and lower weakness, dysarthria, increased falls and gait instability after 650 U of onaBTX-A [33]. No serious AEs were reported by studies included in the present review and only transient and mild side effects were described [12,38,39]. Among the studies that treated post-stroke patients, Mancini et al. reported that some subjects receiving high dose onaBTX-A showed weakness of the treated limb, flu-like syndromes and oedema of the injected leg enduring for more than 4 weeks [27]. The large sample study of 155 studies of incoBTX-A escalating doses reported that in total, 36.1% (56/155), 37.5% (57/152), and 25.7% (36/140) of patients reported AEs in cycles 1 (400 U), 2 (600 U) and 3 (800 U), respectively, and no differences were observed between groups. Furthermore, there was no increased incidence of AEs with increasing doses or repeated injections [38]. Chiu et al. reported that 13 (19%) patients had AEs at 1 year, and that the most common of these was bruising. However, no patients suffered from serious AEs and only one patient discontinued injection due AEs [12]. Ianieri et al., considering all injections, described only four cases (3.3%) of excessive local muscle weakness and two cases (1.6%) of transient generalised weakness lasting 20 and 10 days, respectively (39). Only 2 (4%) subjects of group C (700–1000 U) complained of transient generalised muscle weakness. Of studies that enrolled mixed samples, mild AEs consisting of generalised weakness (12%), feeling of residual urine (10%), constipation (9%) and blurred vision (8%) were observed by Dressler et al., but these disturbances were attributed to underlying neurological conditions and not to the effect of BTX-A [11]. The study by Kirshblum et al. analysed if adverse events increased when injecting higher doses of BTX-A, and if onaBTX-A or incoBTX-A differed in adverse event rates [29]. They reported that AEs did not increase until doses of BTX-A exceeded 600 U. AEs were observed in 7 subjects (5.6%), 4 (2.2%) and 16 (2.6%) treated by ≤400U; 400–600 U and >600 U of BTX-A, respectively. Higher doses > 600 units were associated with an increased risk of complications (OR 2.98, CI 1.14–7.78). However, the authors calculated that the number needed to harm for most related complications exceeded 80 and suggested that the benefits of high dose BTX-A may outweigh the risks. No statistical difference in AE rate was detected between incoBTX-A and onaBTX-A.

Other potential complications of injecting BTX-A include the occurrence of neutralising antibodies (Nab) [72] that counteract the effect of the neurotoxin. This phenomenon has represented an important consideration, limiting the use of increased doses of BTX. Repeated injections and BTX-A high doses have been considered potential risks in promoting Nab [73,74]. However, no studies included in the present review reported this complication, even if almost all of them did not ascertain Nab by laboratory test; the exception is the study by Wissel et al. that ascertained Nab occurrence with escalating incoBTX-A high doses. Blood samples were taken for antibody and laboratory assessments, but none of the patients developed Nab or a secondary response.

## 3. Discussion

Studies about the role of high BTX-A injections in treating spasticity following UMNS are scant and there remains insufficient evidence to recommend routine use in clinical practice. The literature search identified 13 studies, according to the selection criteria, and most of these investigated the effect and safety of BTX-A high doses in post-stroke spasticity, but total number of injected subjects did not exceed 300. The studies were variable in method design, sample size, sample type, aim, outcome measures and only two were RCTs. Despite the poor quality of studies, high BTX-A dosage up 840 U were efficacious and safe in reducing multilevel spasticity or in treating both UL and LL in same session and certain patients with spasticity might benefit from high dose BTX-A, particularly post-stroke subjects. BTX-A use in treating spasticity due to brain injury was investigated in a few studies, all of which enrolled mixed samples. Of these, only two studies reported the number of injected subjects [38,43], but the total number sample was very small since only 27 patients were identified. Although, spasticity is a common and troublesome complication of MS and SCI, none of the studies investigated the effect and safety of high dose BTX-A in treating spasticity of subjects suffering from these diseases. One study enrolled a sample in which subjects with MS and SCI were also injected with high onaBTX-A or incoBTX-A injections, but the number of those was not reported, and neither were the outcomes according to spasticity aetiology; therefore, no remark could be made. To date, high doses of BTX-A have been predominantly used in treating post-stroke spasticity. The explanation of limited use only for stroke may be attributed to several reasons: 1) the long-term experience and familiarisation in injecting post-stroke subjects; 2) the complexity of neurological features with coexistent symptoms including apraxia, ataxia, fatigue and weakness other than spasticity such as in MS; 3) the availability of licensed BTX-A formulations and authorisation for injecting in local post-stroke spasticity; 4) multi-level spasticity involving upper and lower limb requiring treatment and doses that might exceed dosage currently approved for each treatment session; 5) more definite objectives in the context of rehabilitative processes to reach specific functional outcomes by neurotoxin injections.

In regard to the last points, post-stroke patients can display a wide variety of features and spastic muscle patterns; herein the opportunity to adapt the dosage of therapeutic BTX injections accordingly and the need for more tailored treatment options and flexibility in doses to inject for sessions [10]. Indeed, surveys conducted in Europe and North America showed that physicians stated that they would inject higher doses of BTX for the treatment of spasticity, if indicated [75]. Furthermore, according to their opinions, the outcome and satisfaction of post-stroke patients could be improved by 75.8% and 78.8%, respectively, by higher BTX-A doses than labelled [9]. Although the reduction of spasticity is widely demonstrated with BTX-A treatment, its impact on the improvement of functional outcomes remains debated and controversial. As aforementioned, a reduction in spasticity has a key role to reach functional outcomes with the intent to improve mobility and dexterity, achieve physiological movement patterns, reduce pain, facilitate nursing measures and avoid complications. However, some reviews have suggested that that some oriented-focused movements unequivocally improve after reduced spasticity as a result of BTX-A treatment, in particular in the upper limbs [76,77]. This finding was also observed after high-dose BTX-A injections [35,36,38].

Weakness is one the main complication and concern in treating spasticity by BTX-A. In this respect, when reducing muscular tone, it has to be taken into account that muscle hyperactivity may have positive functional aspects, such as stabilising paretic limbs. Moreover, treating muscle hyperactivity does not always improve coexisting paresis. Indeed, treatment plans must consider a trade-off between a reduction in spastic hypertonia and the preservation of residual motor function [78]. In the present review, doses of BTX-A above 600 U produced transient weakness and local mild AEs in almost studies, but no serious AEs were observed. Kirshblum et al. reported an increase in AEs but suggested that the benefits of high dose BTX-A may outweigh the risks of AEs and may be clinically acceptable in certain patients [29]. The present review showed that among BTX-A formulations, onaBTX-A and incoBTX-A were used and of these, incoBTX-A was mainly injected though not authorised for LLS. IncoBTX-A is a purified neurotoxin without complexing proteins, and it may be argued that this formulation lacking accessory proteins has been considered to be less likely to lead to the development of Nab. Thus, patients are more able to adapt to high doses and this lends itself towards flexible dosage use in treating spasticity. However, high doses of both incoBTX-A and onaBTX-A resulted in efficacy and safety at doses between 600–840 U [11,12,28,29,34,35,36,37,38,39,43] and dosages up to 1200 U [39] were injected without occurrence of Nab or serious AEs.

An important and widely debated issue concerns the cost-utility of BTX-A compared to the benefit gained, particularly in treating post-stroke spasticity. Investigations have demonstrated that aboBTX-A adjunct to rehabilitation produces a higher number of quality-adjusted life years compared to rehabilitation alone either in treating ULS [79] or post-stroke spasticity [80], and this therapeutic approach might be a cost-effective healthcare program for treating these patients [80]. In the present review, none of the studies included reported the cost-effectiveness of high BTX-A injection in treating spasticity, regardless of the aetiology.

To date, many questions remain unsolved regarding the use of high-dose BTX-A in treating adults with spasticity following CNS damage. Therefore, future well-designed studies should be planned to address the following issues:What subjects and spastic patients are suitable to undergo high-dose BTX-A treatment. This is particularly the case if the same benefit is not achievable with the recommended dosage under expert and proper guidance in muscle injection.If both UL and LL or multilevel segments of body sites should be injected for each injection cycle.The maximum doses for muscles and what is the maximum dosage to inject within the same session.The interval of injection, the length of the duration effect and long-term follow-up.The specific objectives and functional goals within rehabilitation processes to improve dexterity and quality of life.The cost-effectiveness of high dose BTX-A.

## 4. Conclusions

The global dosage injected per muscle or multiple muscle groups has progressively increased over time and both onaBTX-A and incoBTX-A high doses have been injected in treating spasticity following CNS damage, particularly in post-stroke patients. We identified few investigations with small numbers of subjects that investigated the treatment of spasticity following brain injury, and no studies were retrieved that ascertained the efficacy and safety of high-dose BTX-A in reducing spasticity following MS and SCI. The studies investigating high dose BTX-A in post-stroke spasticity had variable method designs and there was insufficient evidence to recommend routine high dose incoBTX-A and onaBTX-A use in clinical practice. However, dosages of these neurotoxins up 840 U were efficacious and had a good safety profile without serious adverse events. In selected patients, the benefits of high-dose BTX-A may outweigh the risks of AEs and may be clinically acceptable. Several issues should be addressed by proper well-designed, planned studies; meanwhile, the functional benefit compared to risks should be taken into account when using high-dose BTX-A.

## 5. Materials and Methods

A search of relevant studies was conducted in MEDLINE/PubMed, the Cochrane Central Register of Controlled Trials, CINAHL, EMBASE, Web of Science and Scopus databases. We included English language reports from the international literature published from January 1989 to February 2020. Search terms varied slightly across databases but included “botulinum toxin”, “botulinum toxin type A”, “spasticity”, “botulinum toxin high doses” “botulinum toxin high dosage”, “post-stroke spasticity”, “upper and lower limb spasticity”, “brain injury”, “multiple sclerosis”, “spinal cord injury” and “adverse events”. We search for “randomized controlled trial” as either MeSH terms, keywords or subject headings. Related terms were combined using the Boolean “OR” and “AND”. Search limits included only adults. We did not include congress abstracts/posters or articles that were not peer-reviewed. Studies were included if: (1) they were homogenous or mixed samples studies enrolling subjects who suffered from spasticity following stroke, brain injury, multiple sclerosis and spinal cord injury; 2) the sample size included four or more subjects; 3) BTX-A formulations licensed by the USA and European authorities were used; 4) doses of BTX-A higher 600 U were injected, regardless of the aim of the study; 5) high doses of BTX-A were injected alone or combined with adjunctive interventions. Studies that investigated high BTX-A dosage in children with spasticity were excluded.

## Figures and Tables

**Table 1 toxins-12-00315-t001:** Studies investigating BTX-A high doses in patients with post-stroke spasticity.

Study/ Year	Design	Sample/ Patients/Sex	BTX-Type and Doses/Guidance/PT	Foll-Up	Measures	Adverse Event	Drop-Out; Lost	Outcome
Santamato et al. 2013 [35]	open label prospective	N = 25 12F, 8 M; age (range 45–71 yrs)	incobotulinumtoxinA (Xeomin) 840 U (ranged from 750 to 840 U) in both UL and LL; UL muscles received a dosage of maximum 540 U; 340 U was administered in LL (range 250–340 U); US; stretching exercises of the muscles injected for 10 days	3 mo	AS; DAS; GATR; VAS	no adverse event	-	improvement in disability, spasticity-related pain, and muscle tone. Significant decrease evaluated after 30 and 90 days from the treatment (*p* < 0.05)
Invernizzi et al. 2014 [37]	Case control	N = 11; 5M, 6 F; age from 44 to 72 yrs	incobotulinumtoxinA (Xeomin) higher 600 U; 12 U/kg (range 600–800); NR		AS > 2; ECG for HRV (RR interval)	no effect on RR interval	-	N/A
Baricich et al. 2015 [34]	cohort; retrospective	N = 26; M 13, F 13; mean age 54.7 ± 11.6	onabotulinumtoxinA (Botox) 600 IU; 13 pts > 700 IU; mean dose 676.9 ± 86.3 IU; US; 23 pts were treated at both upper and lower limb; electrical stimulation and stretching of injected muscles, strengthening exercise, gait training	3 mo	MAS; DAS; GAE	no adverse event	-	significant reduction of spasticity (*p* < 0.0001)
Mancini et al. 2015 [27]	randomised, double-blind, dose-ranging study	N = 45 pts; N = 15 pts with onaBTX-A high dosage; (M 8, F 7) mean age 63.2 ± 10.1	onabotulinumtoxinA (Botox) 540 ± 124.2 U; EMG	4 mo	MRC; MAS; VAS GT; GV;	prolonged weakness of the treated limb, flu-like syndrome and oedema of the injected leg, in some patients enduring for more than 4 weeks	-	prolonged effect of BTX on spasticity, GV, gait function, pain and presence of clonus
Santamato et al. 2017 [36]	open label prospective	25 pts; 20 (12F, 8 M); mean age 60.8 ± 7.8	incobotulinumtoxinA (Xeomin); 830 U (ranged from 750 U to 830 U) in both upper and lower limb; US; upper limb muscles received a dosage of maximum 560 U; a dosage of maximum 460 U was administrated into lower limbs (ranged from 260 U to 460 U); stretching exercises of the muscles injected for 10 days	2 yrs	AS; DAS; GATR	no adverse event	5 pts	improvements as assessed on clinical scales for spasticity (AS), disability (DAS) and global assessment of treatment response (GATR)
Wissel et al. 2017 [38]	prospective, single-arm, dose-titration study	mixed sample N = 155 pts M 104; F 51; mean age 53.7 ± 13.1 N= 132 with stroke; N= 23 other causes^	incobotulinumtoxinA (Xeomin) 400 to 800 IU	36–48 wks	AS; REPAS; GAS; Investigators’ global assessment of tolerability; Investigators’ and patients’ global assessments of efficacy	no treatment-related serious adverse event (AE) occurredN= 5 pts;The most frequent AEs overall were falls (7.7%), nasopharyngitis, arthralgia, and diarrhea (6.5% each)	18 pts	dosage up to 800 U was safe and was associated with increased treatment efficacy, improved muscle tone, and goal attainment
Baricich et al 2017 [28]	single blind randomized controlled crossover study design	10 pts; 7 M, 3 F; age 69 ± 10.5	N = 5 onabotulinumtoxinA (Botox) 600 U (670 ±83.67); N = 5 IncobotulinumtoxinA (Xeomin) (660 ± 89.44); doses below 12 units/Kg	N/A	AS; BI; MI; and FAC score	-	N/A	no influence on the cardiovascular activity of the autonomic nervous system in chronic hemiplegic spastic stroke survivors.
Ianieri et al. 2018 [39]	retrospective	mixed sample° N = 120 N= 58 M 28, F 22; mean age 66± 3.2	incobotulinutoxinA N = 58 received 700-1000 U (from 775.65 ± 30.45 to 986.65 ± 13.67); NR; stretching of injected muscles, active and passive limb mobilization, walking training, and global muscle strengthening, daily for the first 30 days after injection	2 yrs	AS; FIM; MyotonPRO	Dysphagia (2%), local (4%) and general muscle weakness (4%);	-	reduction of spasticity with mild transient AEs consisting in local weakness (3.3%)and generalized weakness (4%) in group injected by higher dosage (from 700 U to 1000 U)
Chiu SY et al. 2020 [12]	retrospective	mixed sample N = 68 pts F 43, M 25; N = 24 with spasticity*, remaining affected by dystonia**	onabotulinumtoxinA (Botox) > 400 U receiving doses up to 800 U; NR	12 mo up 86 mo	7-point Clinical Global Impression Scale (CGIS)	Ten patients (15%) reported adverse effects (AEs) at the first follow-up;13 patients (19%) reported AEs at 1 year. The most common AE reported was bruising	38 pts at last fol.-up	all patients reported benefit after first treatment (8.8 weeks ± 3.1)

Legend: N/A = not applicable; NR = not reported; PT = Physical therapy; EMG = electromyography; US = ultrasonographic guide; AS = Ashworth scale; BI = Barthel Index; DAS = disability assessment scale; GAE = global assessment of efficacy; GV = gait velocity; GAS = goal attainment scale; GATR = global assessment of treatment response; GOS = Glasgow outcome scale; FAC = Functional ambulation category score; HRV = heart rate variability; FIM = Functional independence measure; MAS = modified Ashworth scale; MI = Motricity Index; MRC = Medical Research Council scale; REPAS = Resistance to Passive Movement Scale; VAS = visual analogue scale; VAS GT = Visual Analogue Scale for Gait Function. ^brain injury, cerebral palsy, brain tumor; °spasticity due to stroke, traumatic brain injury, multiple sclerosis, spinal cord injury; *spasticity: common etiologies included stroke, traumatic brain injury, and cerebral palsy; ** primary dystonia, idiopathic Parkinson’s disease, or atypical parkinsonian syndrome.

**Table 2 toxins-12-00315-t002:** Studies that enrolled mixed samples. Subjects with dystonia and spasticity following disorders of CNS who underwent BTX- A high doses.

Study	Design	Patients/Sex	BTX-Type and Doses/Guidance/PT	Follow-Up	Measures	Adverse Event	Outcome
Dressler et al. 2009 [69]	open label prospective	N = 236 pts; N = 84 pts with spasticity^&^	incobotulinumtoxinA at 450.5 ± 177.1 U;maximum botulinum toxin dose applied was 840 U;	3 yrs	NR	none of the patients experienced systemic adverse effects, neither motor nor autonomic ones.	no subjective or objective differences detectable compared to Botox previously injected
Intiso et al. 2014 [43]	open label prospective	N = 22 pts; mean age 38.1 ± 13.7 years; N = 16 with BI; N = 6 with CP	incobotulinumtoxinA up to 840 U; US	16 wks	MAS, MRC, VAS, FAT, GOS, BI	hematoma (2 pts); muscle weakness and reduction of active motility of the injected arm (1 pt)	high-dose BTX-A injections were effective and safe in reducing spasticity. Significant reduction of the pain was also observed
Dressler et al. 2015 [11]	prospective non interventionalrandomized study	N = 100 pts; N = 46 pts with dystonia N = 54 pts with spasticity^&^; 33 M, 21 F; mean age 56.1 ± 14.7 years	incobotulinumtoxinA 612.6 ± 176.5 (min 400, max 1.200) U; EMG or US	NR	STQ;	generalized weakness (12%);feeling of residual urine (10%);constipation (9); blurred vision (8%), attributed to underlying neurological condition	doses > 400 U and up to 1200 U without detectable systemic toxicity were used safely. No developed Nab
Kirshblum S et al. 2020 [29]	retrospective	N = 342 pts F = 190; M = 152 Mean age 53.1 ± 16.3 yrs including cervical dystonia and spasticity^&^	onabotulinumtoxinA or incobotulinumtoxinAN = 42 pts (14%) received > 600 U	3 yrs	-	subjests receiving > 600 Uadverse events (5.6%);weakness (4%);dysphagia (1.6%)	increased risk of adverse events associated with BTX-A doses higher than 600 U (OR 2.98, CI 1.14–7.78). There was no difference in adverse events between onabotulinumtoxinA or incobotulinumtoxinA

Legend: NR = not reported; BI = brain injury; CNS= central nervous system; EMG = electromyography; US = ultrasound; GOS = Glasgow Outcome scale; FAT = Frenchay Arm Test; MAS = modified Ashworth scale; MRC = Medical Research Council scale; Nab = neutralizing antibodies; STQ = Systemic toxicity questionnaire; VAS = visual analogue scale ^&^ Spasticity etiology was not specified;.

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
