# Peer review of "High Dosage of Botulinum Toxin Type A in Adult Subjects with Spasticity Following Acquired Central Nervous System Damage: Where Are We at?"

_toxins, 2020, doi:10.3390/toxins12050315_

Round 1
Reviewer 1 Report
This is a very interesting and valuable work, but I had a difficult time following the manuscript due to the writing style. I would suggest major revision to improve readability. In addition, during the revision I would encourage editing to streamline the text and avoid repetition of phrases.
The results section seemed to overlap a great deal with the discussion- there was a significant amount of discussion/conjecture rather than strict presentation of the data. I did appreciate the organization of the results section by diagnosis and toxin. I would only pare down the text and move the discussion to the discussion section itself.
Author Response
Response to Reviewer 1
We do thank Reviewer n. 1 for her/his interest in our research and for thoughtful words.
Her/his suggestions were added to text in the section BTX-A
Page 2, lines 75-83
Botulinum neurotoxin (BTX), so called botulinum toxin, is a protein synthesized by the gram-negative anaerobic bacteria Clostridium botulinum. It occurs in the form of seven serologically distinct types. Each of which comprises numerous isoforms. Most of the research available in the medical literature regards the neurotoxin type A (BTX-A). The classic site of the impact of the toxin is the protein complexes of the presynaptic membrane and synaptic vesicles, implementing the acetylcholine exocytosis. Type A botulinum toxin can be effective in the treatment of drug-resistant migraine. It is also widely used in aesthetic medicine for the correction of age-related changes in muscle tension, but also otherwise for the treatment of painful bladder, chronic myalgia, blepharospasm and some kinds of neuralgia
Reviewer 2 Report
Botulinum neurotoxin (BoNT, BTX), so called botulinum toxin, is a protein synthesized by the gram-negative anaerobic bacteria Clostridium botulinum. It occurs in the form of seven serologically distinct types. Each of which comprises numerous isoforms. Most of the research available in the medical literature regards the neurotoxin type A (BTX-A). The classic site of the impact of the toxin is the protein complexes of the presynaptic membrane and synaptic vesicles, implementing the acetylcholine exocytosis. Type A botulinum toxin can be effective in the treatment of drug-resistant migraine. It is also widely used in aesthetic medicine for the correction of age-related changes in muscle tension, but also otherwise for the treatment of painful bladder, chronic myalgia, blepharospasm and some kinds of neuralgia.
This very comprehensive review is an accurate, well-written and wide-ranged source of current data dealing with the application of BTXs in the treatment of post stroke spasticity, multiple sclerosis and brain injury. The topic is particularly timely and of potential interest for the readers, especially for neurologists. After precise and critical analysis, Authors conclude, that there is no sufficient evidence to recommend BTXs injection as a standard procedure, however the neurotoxin is often efficacious and had a satisfactory safety profile in the clinical practice. This work is an important and valuable contribution to the field.
Author Response
Response to Reviewer 2
We do thank Reviewer n. 2 for her/his insightful comments and observations, which were considered in the revised version of our manuscript.
C 1. This is a very interesting and valuable work, but I had a difficult time following the manuscript due to the writing style. I would suggest major revision to improve readability. In addition, during the revision I would encourage editing to streamline the text and avoid repetition of phrases.
Reply: we revised and deleted some redundant sentences through the text, in particular for the section “discussion” that was globally reworded and simplified. The section “stroke” was in part reworded. Before submitting the paper for peer review process, English language was revised by Proof-Reading- Service.com Ltd, Devonshire Business Centre, Work Road, Letchworth Garden City, Hertfordshire (United Kingdom).
Section Stroke
Line 100-101: deleted.
Lines 134-135: deleted.
Section Multiple Sclerosis
Lines 286-289: deleted
Line 322: incoBTX-A or onaBTX-A doses higher 600 U in a large sample of 342…. was deleted
Line 366: due to their reporting of fewer than four subjects.. deleted
C 2. The results section seemed to overlap a great deal with the discussion- there was a significant amount of discussion/conjecture rather than strict presentation of the data. I did appreciate the organization of the results section by diagnosis and toxin. I would only pare down the text and move the discussion to the discussion section itself.
Reply: we agree with reviewer’s comments that some parts of results overlap discussion, but some important considerations about the topic were intentionally reported in the section of results to avoid a too long discussion. The strict presentation of the data is essential, but the hypotheses to explain those should be carried out. The aim of paper was to ascertain evidence about the use of BTX-A high doses in treating spasticity following several diseases affecting CNS in subjects requiring rehabilitation. In this regard, the reasons and significance in using this particular therapeutic strategy should be considered and discussed. However, according with reviewer’s comments, we reworded and simplified the discussion.
Round 2
Reviewer 1 Report
Excellent revisions. Again, this review paper is timely and should be of great interest to the readership.